# The Role of Organoids in Advancing Colorectal Cancer Research: Insights and Future Directions

**DOI:** 10.3390/cancers17132129

**Published:** 2025-06-25

**Authors:** Zahra Heydari, Rex Devasahayam Arokia Balaya, Gobinda Sarkar, Lisa Boardman

**Affiliations:** 1Department of Gastroenterology and Hepatology, Mayo Clinic, Rochester, MN 55905, USA; devasahayamarokiabalaya.rex@mayo.edu (R.D.A.B.); sarkar.gobinda@mayo.edu (G.S.); 2Department of Laboratory Medicine and Pathology, Mayo Clinic, Rochester, MN 55905, USA

**Keywords:** organoids, colorectal cancer, 3D models, patient-derived organoids, personalized medicine

## Abstract

This review focuses on organoids, which are three-dimensional structures that mimic human organs. The authors aim to show how organoids can improve our understanding of colorectal cancer, which is a significant global health issue. Organoids derived from patient tumors replicate the complex interactions within the living tissue, therefore providing a more realistic model of the tumor. This allows researchers to study the disease more accurately than with traditional two-dimensional cell cultures. This approach enhances drug testing and personalized treatment strategies, which can lead to improved patient outcomes.

## 1. Introduction

The global incidence of colorectal cancer (CRC) has been consistently rising. CRC is the third most diagnosed cancer globally, and second ranked cause of cancer-related deaths [1]. There are many causes which have led to the rise in CRC involving a combination of genetic and environmental factors. These factors increase an individual’s susceptibility to the disease [2]. Despite significant advancements in understanding CRC at the molecular and genetic levels, which have led to improved treatments and survival rates, challenges remain. The increasing incidence of early-onset CRC and regional variations in survival highlight ongoing deficits in prevention and treatment strategies [3,4]. Furthermore, drug resistance remains a significant barrier to effective treatment outcomes, while personalized and tailored treatments remain largely inaccessible to most cancer patients [5].

Current research and antitumor drug development generally utilize traditional tumor models, which include two-dimensional (2D) cultures and patient-derived tumor xenografts (PDXs). Two-dimensional models have numerous limitations, such as an inability to accurately replicate the complex interactions within living tissues, lack of accurate cell–cell and cell–extracellular matrix (ECM) interactions and loss of critical tumor characteristics during cell passage. Moreover, PDXs in mice can undergo significant changes, differing from their human counterparts. The high costs, prolonged developmental timelines, and limited efficacy rates in current tumor models, highlight the vital need to improve novel methodologies [6].

The investigation of the complex multicellular components of tumors has traditionally relied on in vivo models. While vivo models have advantages over in vitro studies, they are not without limitations. These limitations include the high costs associated with animal models, the necessity for large sample sizes for time-point experiments, challenges in cell imaging sensitivity in vivo, and ethical concerns regarding animal use. Additionally, interspecies differences can impede the translational potential of therapeutic interventions from animal models to human patients [7,8].

The concept of organoids has gained significant traction in recent years as an innovative approach to studying the complex multicellular components of tumors. The first successful establishment of an organoid was achieved by H. Clevers in 2009 [9]. Since then, organoids have become a prominent area of research, offering promising solutions for various biological applications. These 3D structures mimic the organization and functionality of native organs, providing a more accurate representation of cellular and extracellular matrix interactions that occur in vivo. The potential applications of tumor organoids are extensive, encompassing the investigation of disease mechanisms, tissue transplantation in regenerative medicine, and high-throughput drug screening. Organoids hold considerable promise for the development of targeted therapies tailored to individual patients, enabling personalized drug combinations and more effective treatment strategies [10,11] (Figure 1).

The development of CRC organoid models has undergone significant progress over the years. The incorporation of stromal or immune cells into co-culture 3D models has provided valuable insights into the roles these cells play within the TME [12,13].

The TME is a complex and dynamic ecosystem that encompasses an assortment of cell types, which include mesenchymal stromal cells (MSCs), tumor-infiltrating immune cells, cancer-associated fibroblasts (CAFs), ECM, and vascular components [14,15].

The MSCs in the TME of CRC have a crucial role in facilitating tumor growth and progression. These cells, which include fibroblasts, smooth muscle cells, and pericytes, can become activated to differentiate into CAFs. Distinct subtypes of CAFs contribute to tumor progression through various mechanisms. For instance, inflammatory CAFs (iCAFs) play a key role in establishing an immunosuppressive microenvironment that promotes tumor survival and resistance to conventional therapies. They do so by secreting cytokines and chemokines that modulate immune cell recruitment and function [16,17].

The immune infiltrate in CRC is composed of diverse subtypes of immune cells, including monocytes, T-cells, neutrophils, mast cells, natural killer (NK) cells, and endothelial cells. Among these, tumor-associated macrophages (TAMs) are particularly influential within the TME. The presence of TAMs has been correlated with poor prognosis in over 80% of human cancers [18]. These macrophages promote tumor growth by enhancing angiogenesis, inducing genomic instability, and suppressing T-cell-mediated anti-tumor immunity. TAMs contribute to chemoresistance by creating an environment that favors tumor cell survival [19].

Additionally, the ECM plays a crucial role in CRC by providing structural support for cell proliferation, growth, and migration [20]. The ECM’s composition and stiffness change dynamically throughout CRC development, influencing cancer progression and potentially reducing drug delivery and promoting resistance. Developing ECM-based models or models that incorporate ECM components could improve the accuracy of in vitro representations of the CRC tumor microenvironment. This can facilitate the development of novel treatments to target immunosuppressive effects [21].

Innovative technologies such as microfluidic systems and 3D bioprinting offer promising avenues for integration with organoid technologies. This serves to enhance viability and functionality over the long term, while more accurately mimicking the characteristics of the original tissues beyond the co-culture with supportive cells [22,23]. Importantly, the gut microbiota has also emerged as a key modulator of CRC progression, influencing tumor behavior, immune interactions, and therapeutic responses. Recent advances in organoid models offer new opportunities to model these host–microbiome interactions in vitro, thereby enhancing their relevance for translational research.

A systematic review showed that organoid-based prediction of treatment response has a positive predictive value (PPV) of 68% and a negative predictive value (NPV) of 78%. These values suggest that patient-derived organoid (PDOs) can significantly outperform empirically guided treatment selection, providing a more personalized approach to cancer therapy [24]. In contrast to PDXs, the success rate of tumor organoid establishment is significantly higher, ranging from 50% to 90% compared to the 10% to 30% success rates typically observed in PDX models [25].

Recent systematic reviews have significantly expanded our understanding of CRC organoid models, particularly their role in translational research. A 2024 review by Zhang et al. emphasized how PDOs have become instrumental in optimizing combination therapies and high-throughput drug screening protocols, helping to bridge the gap between preclinical findings and clinical applications [26]. Additionally, Wang et al. provided a comprehensive overview of CRC organoids integrated with immune cells and stromal components, underscoring the enhanced physiological relevance of these models for studying tumor–immune interactions and resistance mechanisms [27]. A 2024 study by Esposito et al. developed a patient-derived immunity–organoid platform to model responses to immune checkpoint inhibitors (ICIs) in CRC. This platform enabled the identification of cancer-specific tissue markers associated with ICI resistance, particularly in microsatellite stable (MSS) CRCs. Notably, the study highlighted REG4 as a new predictive biomarker of immunotherapy resistance. Knocking out REG4 in resistant organoids restored immune sensitivity and triggered T-cell-mediated apoptosis, underscoring its potential as a therapeutic target to overcome immune resistance in CRC [28]. A 2025 review by Wu et al. highlights the growing importance of organoid models in CRC research. The authors emphasize the ability of an organoid to mimic organ structure and function, making them valuable tools for studying CRC biology. The review covers both adult and pluripotent stem cell-derived organoids and explores their applications in basic research, drug development, personalized therapy prediction, and regenerative medicine [29].

This review summarizes recent advances in the use of CRC organoids for disease modeling, drug screening, and personalized medicine, and outlines current challenges including standardization, scalability, and clinical translation. It will review foundational research and preclinical data, while also highlighting the most promising technologies, including integration of microbiota, that can be applied to organoids to enhance their functionality in CRC.

## 2. Organoid Platforms to Study Host–Microbiota Crosstalk in CRC

Understanding host–microbiome interactions is critical for elucidating the mechanisms underlying CRC tumorigenesis, as the gut microbiota plays a pivotal role in shaping the tumor microenvironment. These microorganisms influence CRC development through immune modulation and the production of metabolic byproducts that affect cellular signaling pathways. Recent advancements in PDO models offer physiologically relevant systems to investigate these complex interactions in vitro, enabling researchers to model the cumulative effects of microbial metabolites and immune responses more accurately. Such approaches not only enhance our understanding of CRC progression but also pave the way for microbiota-informed therapeutic strategies. Specific bacteria, such as *Fusobacterium nucleatum*, genotoxic *pks+ Escherichia coli*, and enterotoxigenic *Bacteroides fragilis* (ETBF), have been consistently linked to CRC [30].

Other genera including *Porphyromonas*, *Parvimonas*, *Streptococcus*, *Gemella*, and *Prevotella*, along with members of the Clostridiales order, are also associated with CRC development. Fungi also play a role, with increased abundance of opportunistic fungi like *Malassezia* spp. and *Candida albicans* in CRC patients [31]. Eukaryotic viruses such as Papillomaviridae or Polyomaviridae are increasingly considered as possible contributors in the development of CRC. While associations between the gut microbiota and CRC show promise as a diagnostic tool, it is crucial to determine which microbes contribute causally to CRC development using various biological systems, in order to develop strategies to combat relevant CRC processes mediated by microbes [32,33].

The gut microbiota metabolizes undigested dietary compounds, producing a diverse array of metabolites that can either promote or inhibit tumorigenesis. These metabolites can influence the development of CRC by modulating the host’s immune response, inducing inflammation, and altering the function of gut epithelial barrier. For example, certain metabolites such as short-chain fatty acids (SCFAs) can inhibit tumorigenesis by promoting the growth of beneficial bacteria and enhancing the immune response, while others like secondary bile acids can promote tumorigenesis by inducing inflammation and DNA damage. The metabolic products of the gut microbiota also interact with the host’s metabolic pathways, influencing the expression of genes engaged in tumorigenesis and CRC development [34,35].

On the other hand, the colorectal epithelium is a crucial component of the gut microbiota, influencing the balance of the ecosystem. Despite the importance of this interface, the understanding of host–microbe interactions remains limited due to the lack of suitable coculture systems. Organoids have emerged as a valuable tool for modeling host–microbe interactions. Researchers have used organoids to study a variety of microbes, including bacteria, viruses, and parasites, that are difficult to culture using 2D cultures or animal models [36].

Colonic epithelial cells can be cultured as organoids under normoxic conditions; however, the conflicting oxygen requirements of epithelial cells and anaerobic gut bacteria present a challenge for stable co-culture. Recent technologies, such as microfluidic systems, are beginning to overcome these limitations by enabling controlled oxygen gradients, which facilitate more realistic modeling of host–microbiome interactions [37].

Although the co-culture of CRC organoids with microbiota is limited, several studies have examined the interactions of microbial species within the gut and their potential impact on CRC development and progression [33,38,39]. While the specific microbial composition associated with CRC organoids and how it may differ from healthy colorectal tissues has not been fully characterized, researchers are actively investigating this area. Additionally, studies are examining the various ways in which the gut microbiota can influence the growth, differentiation, and other key characteristics of CRC organoids, such as through the production of metabolites, the modulation of immune responses, or the alteration of signaling pathways [40,41,42]. By deepening the understanding of the complex interaction between CRC organoids and the gut microbiota, researchers can gain valuable insights into the role of the microbiome in CRC and potentially identify new avenues for improving cancer prevention, diagnosis, and treatment [43] (Figure 2).

Organoid microinjections allow researchers to introduce living bacteria and parasites directly into the organoid lumen, capitalizing on the hypoxic environment that supports the growth of anaerobic bacteria such as *E. coli*, *Fusobacterium nucleatum*, and *Clostridioides difficile*. This method facilitates the establishment of multi-day in vitro co-cultures, enabling the investigation of microbial interactions with the gut epithelium. This closely mimics their natural environment in vivo. By applying non-permeable antibiotics post-microinjection, this model can maintain luminal bacteria while eliminating external contaminants [44]. To preserve luminal microbes while eliminating external contamination, non-permeable antibiotics such as gentamicin are applied to the external culture medium following microinjection. These antibiotics act extracellularly and, at appropriate concentrations, effectively control contamination without impairing epithelial viability or disrupting organoid structure [45].

Using human adult stem cell-derived organoids, in a 2021 study, Clevers and colleagues established protocols for microbe–epithelium co-cultures, which exhibit a cystic structure that mimics the gut lumen. These comprehensive protocols build upon previous microinjection techniques developed for various organoid systems and pathogens. In addition to the microinjection method, the protocols also include alternatives that do not require injections, such as viral exposure or 2D cocultures in Transwell plates. These protocols have broad applications for studying interactions between microbe–epithelium in contexts such as infectious diseases, cancer, and homeostasis. These protocols are particularly well-designed to investigate the impacts of individual microbial species while allowing for modifications to study combinations of different microorganisms. Employing organoids derived from adult stem cells ensures an accurate representation of wild-type epithelial cell types. This enables detailed characterization of target cells for viral or bacterial infections under controlled experimental conditions [46].

There are limitations, however, to this approach that need to be addressed in future research. The manual nature of the microinjection process poses challenges for scaling up to high-throughput setups. Additionally, employing a reductionist approach that relies on individual microbial species fails to capture the intricate nature of the gut microbiota or its interactions with the immune system. The current system also lacks certain features of the gastrointestinal tract, the colorectum in particular, including the crypt-villus architecture, a fully developed mucus layer, and dynamic flow conditions. Furthermore, techniques for controlling critical parameters like oxygen and nutrient levels in coculture systems require further investigations [46].

Alternatively, apical-out organoids present a novel approach by reversing the polarity of organoids to expose the apical side without microinjection. One must consider, however, the potential effects on cell phenotype and bacterial overgrowth with this approach. Basal organoid co-cultures, also offer a complementary method to study microbial effects from the basolateral side, primarily focusing on microbial metabolites or toxins rather than live bacteria [44] (Table 1).

Taken together, organoid-based models of host–microbiota interactions offer a transformative platform for understanding the microbial contributions to CRC development and therapy resistance. These models hold potential for uncovering novel microbial biomarkers and tailoring microbiota-targeted interventions that could improve CRC prevention and treatment outcomes.

## 3. Advancements in CRC Organoid Models for Tumor Heterogeneity, Drug Screening, and Co-Culture Systems

PDOs have emerged as an essential tool in CRC research, offering a high-fidelity, patient-specific platform that bridges the gap between in vitro models and clinical relevance. Unlike cancer cell lines, which are widely used for drug screening but lack the spatial architecture, phenotypic heterogeneity, and genetic complexity of primary tumors, PDOs preserve the key molecular and histological features of the tumors from which they are derived. Similarly, while PDX models better retain tumor heterogeneity and allow for in vivo drug testing, they require large tissue samples, have relatively low success rates for engraftment, and involve long experimental timelines. Pluripotent stem cell-derived organoids, on the other hand, are primarily used to model normal colorectal development and do not reflect the mutational landscape of cancer. In contrast, PDOs can be established from small biopsy samples, expanded rapidly, and maintained long-term in vitro while faithfully recapitulating the cellular diversity, mutational spectrum, and treatment response characteristics of patient tumors [56,57].

A 2015 study conducted by Weeber et al. established the feasibility of establishing PDOs from biopsies of patients with metastatic CRC, with a success rate of 71% [58]. This compares favorably with significantly lower success rates observed in other tumor types—for instance, prostate cancer, where organoid culture success rates range from 15 to 20% [59]. Organoids accurately reflect the genetic diversity of the original tumor biopsy, with 90% of somatic mutations maintained between biopsies and organoids from the same patient. While none of the unique mutations identified in this cohort were located in currently druggable driver genes, the high genetic concordance supports the use of PDOs as a reliable ex vivo model. This enables drug screening based on functional sensitivity rather than just genomic targets, highlighting their value in precision oncology, particularly for patients without clearly actionable mutations. Despite these promising figures, several factors contribute to variability in PDO establishment. Biological heterogeneity within tumors, such as sampling differences from distinct tumor regions and ongoing tumor evolution, can lead to discordance between the genomic profiles of organoids and their parent biopsies. Technical limitations also play a role; for example, low sequencing coverage (<25×) may hinder accurate detection of somatic mutations, and low tumor content in biopsy samples, due to stromal contamination or necrosis, can obscure copy number aberrations and reduce mutation allele frequencies below detection thresholds [58].

Janakiraman et al. developed complementary PDX and patient-derived tumor organoid (PDTO) models from pre-treatment rectal cancer specimens to address the lack of disease-specific platforms for rectal cancer. While PDXs successfully mirrored the histology and mutational profiles of the original tumors and replicated response to 5FU/RT therapy, the PDTOs uniquely provided an in vitro system that preserved the heterogeneity of tumor morphology and architecture. Importantly, PDTOs demonstrated a strong correlation with clinical responses to neoadjuvant 5FU/RT therapy and showed differential sensitivity to cetuximab based on *KRAS* status, mirroring patient-specific outcomes [60].

*Integration of Co-Culture Models: Direct vs. Indirect Systems*. Direct and indirect co-culture systems have emerged as a highly effective approach to enhance the functionality of cancer organoids. Direct co-culture systems involve the physical interaction of two or more cell types within the same culture space. This method allows for direct cell-to-cell communication, which can significantly influence cellular behavior and function. Indirect co-culture systems, on the other hand, allow for interactions between different cell types without direct contact. This is achieved using Transwell systems or other separation methods that permit the exchange of soluble factors while keeping the cells physically apart. Recent studies underscore that the choice between direct and indirect co-culture methods profoundly affects experimental outcomes. Direct co-culture facilitates juxtacrine signaling and allows for the exchange of membrane-bound factors, mechanical interactions, and ECM remodeling, all of which are essential in processes such as epithelial–mesenchymal transition (EMT) and collective cell migration. In contrast, indirect co-culture systems rely on paracrine signaling and are ideal for investigating soluble factor-mediated interactions in a controlled microenvironment, though they may lack the complexity of direct physical interactions observed in vivo. Therefore, the selection of co-culture methodology should be tailored to the biological question, as it may influence outcomes related to drug response, EMT, immune modulation, and stromal interactions [61].

In a 2021 study, the investigators addressed the lack of CAFs in PDO models by creating in vitro conditions that allowed the indirect co-culture of human derived CRC PDO and human derived CAFs using hyaluronan–gelatin hydrogels for the 3D matrix. This co-culture method preserved the viability of both the CRC PDO and CAFs, and the presence of the CAFs preserved the proliferation of the CRC PDO and resulted in the CRC PDO expressing biological pathway activity present in the corresponding human tumor samples absent from these same PDO when cultured in the absence of their CAFs. The study highlights the potential of these CRC PDO-CAFs co-culture models for analyzing standard-of-care drugs and realizing personalized cancer medicine [62].

Colon PDOs indirectly co-cultured with CAFs have shown significant upregulation of *ZEB1* and *Vimentin* expression, indicative of partial EMT. While EMT is often linked to increased invasiveness and metastasis, its role can vary depending on tumor context and CAF subtype, reflecting the functional heterogeneity of the tumor stroma [63]. *Vimentin*, a hallmark of the mesenchymal state, supports structural changes that facilitate migration. *ZEB1*, a key transcription factor driving EMT, promotes loss of epithelial characteristics and enhances cell motility and invasiveness. The upregulation of *ZEB1* and *Vimentin* in CRC has been associated with increased invasion and chemoresistance. Additionally, *ZEB1* contributes to chemoresistance by modulating the DNA damage response and activating survival pathways that reduce sensitivity to chemotherapy [64,65,66].

In a study performed in 2023, researchers aimed to elucidate the role of CAFs in mediating chemoresistance and to establish a platform for drug screening. They seeded CRC cells alongside CAFs within a 3D matrix by utilizing a direct co-culture method. The CAFs in co-culture were characterized by high expression of ECM components, glycolysis, enzymes, hypoxia, and immunosuppressive genes. Single-cell RNA sequencing revealed that CAFs induced a partial EMT in a subset of cancer cells, mirroring features of the aggressive CMS4 molecular subtype. Furthermore, the co-culture secretome, rich in immunosuppressive mediators such as TGFβ1, VEGFA, and lactate, potently inhibited T cell proliferation, demonstrating the model’s utility for studying immune evasion [67].

A more recent study by Strating et al., demonstrated that co-culturing CRC organoids with CAFs in an indirect condition. Following treatment with standard chemotherapeutics (5-fluorouracil and oxaliplatin), transcriptomic profiling of co-cultured PDOs revealed significant alterations in gene expression compared to monocultures—particularly in pathways related to IFNα/β signaling and major histocompatibility complex (MHC) class II protein complex assembly. These pathways are critical modulators of immune response and have been implicated in JAK/STAT-mediated drug resistance mechanisms. This study highlights how CAF-organoid interactions reshape the transcriptional landscape of CRC cells in a drug-exposed context, suggesting a dynamic and reciprocal influence between tumor cells and the stromal compartment [68].

*Modeling Immune–Tumor Interactions*. Immune evasion is a known hallmark of cancer, and CRC organoids can indicate how immune–epithelial interactions might be adapted in bowel cancer. NK cells are a type of immune cell that play an essential role in detecting and eliminating cancer cells. However, in the case of CRC, the effectiveness of NK cells can be reduced [43]. In a 2022 study, PDOs of CRC were exposed to conditioned media from NK cells. NK cells are rich in pro-inflammatory cytokines such as tumor necrosis factor (TNF) and interferon-gamma (IFN-γ), known for promoting apoptosis in target cells. Despite the presence of these potent inflammatory signals, the CRC cells exhibited sustained viability. This observation suggests that CRC cells may have developed adaptive mechanisms to resist the cytotoxic effects exerted by NK cells [69]. Although MSI-high CRCs are typically sensitive to IFN-γ, the MSI status of the organoids in that study was not reported. This resistance may reflect immune evasion mechanisms associated with MSS CRCs, which are known to be less responsive to immune-mediated apoptosis.

CRC organoids with mutations in the *APC* gene, which is commonly altered in colorectal cancer, showed decreased expression of the IL-22 receptor and were resistant to treatment with IL-22. This finding highlights another way in which the interactions between immune cells and CRC cells are altered, as IL-22 signaling is known to play a role in regulating inflammation and immune responses in the gut [70].

A 2023 study by Subtil developed a dynamic 3D co-culture structure to model the interactions between patient-derived metastatic colorectal cancer (mCRC) PDOs and monocyte-derived dendritic cells (MoDCs). This system revealed that mCRC organoids significantly modulate the phenotype and function of both immature and mature DCs, including changes in activation marker expression and impaired T cell–stimulating capacity. The co-cultures showed high viability and structural resemblance to patient tissue and allowed for retrieval and functional profiling of DCs. By using PDOs, the model preserves tumor-specific cues that influence immune evasion, offering a more physiologically relevant platform than standard models. However, while powerful, the system focuses on MoDCs and may not fully represent the diversity of in vivo DC subsets, suggesting future expansions to further capture immune heterogeneity [71].

Fang et al. developed a co-culture system combining monocytes, CD8+ T cells, and CRC PDOs to investigate the interactions between immune cells and cancer cells. Specifically, they observed that this autologous co-culture revealed that high levels of SIRT1 in CRC cells enhance macrophage infiltration and M2 polarization, leading to CD8+ T cells dysfunction. At the molecular level, SIRT1 promotes CXCL12 expression by inhibiting p53 acetylation, which attracts CXCR4-expressing monocytes. This leads to increased tumor-associated macrophage infiltration and accumulation, contributing to immunosuppression and impaired CD8+ T cell proliferation and activity [72].

The use of CRC organoids with microsatellite instability (MSI) provides a model to evaluate the efficacy of T-cell-mediated tumor killing by co-culturing peripheral T-cells with PDOs to induce the creation of tumor-specific T-cells that do not distinguish normal organoids from tumor. This structure offers several advantages, including the expansion of reactive tumor-specific T-cells for personalized analysis of their anticancer characteristics, comparison of the relative contribution of different cancer antigens to T-cell killing, and assessment of strategies of adoptive T-cell transfer [73,74]. MSI tumors are characterized by a high tumor mutational burden (TMB) and elevated neoantigen load, which promote robust T cell infiltration and increased responsiveness to PD-1/PD-L1 immune checkpoint blockade [75]. However, despite this immunogenicity, immune evasion mechanisms, particularly through PD-1/PD-L1 signaling, can lead to T cell exhaustion within the tumor microenvironment. To model these interactions, co-culturing peripheral T cells with MSI PDOs enables the generation and functional assessment of tumor-specific T cell responses under immunosuppressive conditions. This co-culture system provides a platform to evaluate the efficacy of checkpoint inhibitors in reversing T cell exhaustion and to model clinical responses to immune checkpoint blockade [73,76].

PDOs co-cultivated with bone marrow stromal cells (BMSCs) induced EMT in CRC cells, leading to an anti-tumor effect through secretion of cytokines that promoted apoptosis and prevented proliferation. Irradiation of the co-culture system further enhanced this effect by cleaving caspase-3 and attenuating PI3K/AKT and ERK signaling pathways [77].

*Potential applications of PDOs in drug screening and personalized medicine*. The potential applications of organoids derived from CRC patients are particularly promising, offering valuable tool for investigating disease mechanisms and evaluating personalized treatment approaches. PDOs have become an asset in the realm of colorectal cancer research, with researchers leveraging these models to advance disease understanding, drug development, and personalized treatment approaches. The process begins by collecting tumor and normal tissue samples from a cohort of CRC patients, which are then used to generate the PDOs. To ensure the reliability of these organoids as representative preclinical tools, different techniques, such as whole-genome sequencing, hematoxylin and eosin staining, and immunohistochemistry, are employed to verify the PDOs maintain similar characteristics as the original patient tumors. The generated organoids are then utilized for drug screening and the development of personalized treatment strategies for individual patients [6].

CRC organoids have emerged as powerful tools to study the genetic and molecular underpinnings of tumor initiation and progression. These organoids can recapitulate the genetic basis of colorectal neoplasia, as proposed by the Fearon and Vogelstein model, in which the accumulation of specific genetic changes, rather than their order, determines the tumor’s biological properties [78]. These models can recapitulate the stepwise accumulation of key genetic alterations, such as mutations in the Wnt, EGFR, TGFβ, and p53 pathways, that drive colorectal carcinogenesis [79,80]. Organoids have also enabled researchers to investigate the role of the tumor microenvironment, including the influence of different fibroblast subtypes and secreted factors, in modulating tumor growth and metastasis [63]. Furthermore, CRC organoids have been utilized to dissect the metabolic phenotypes of cancer stem cells (CSCs) and differentiated cancer cells, providing insights into how non-CSC-derived lactate can promote CSC self-renewal and disease progression [81].

A study by Nguyen et al., 2025 [82] demonstrated that lactate maintains and expands colorectal CSCs by acting as an epigenetic regulator. Lactate increased histone acetylation, particularly H3K27ac, which promoted transcriptional activation of the MYC oncogene, a key driver of CSC self-renewal and plasticity. This effect suppressed CSC differentiation and induced dedifferentiation of non-CSCs back into a proliferative stem-like state, thereby promoting tumor growth and resistance to therapy [82].

As a personalized preclinical model, CRC organoids show significant promise in predicting drug sensitivity and treatment response, though their ability to accurately predict clinical outcomes requires further validation through prospective studies. Organoids derived from metastatic CRC patients have been used to test the sensitivity to standard-of-care chemotherapies, such as 5-fluorouracil, oxaliplatin, and irinotecan, with the results indicating that these models can effectively identify non-responders and prevent patients from undergoing ineffective treatments [83]. While PDOs have shown strong promise in predicting responses to chemotherapeutic agents, their utility in forecasting radiosensitivity is still being explored. The mechanisms of radiation response involve complex DNA damage repair pathways and interactions with the tumor microenvironment, which are not always fully recapitulated in organoid cultures. As such, the predictive accuracy of PDOs may vary between chemotherapy and radiotherapy modalities. Further validation studies are needed to assess the reliability of PDO-based models in guiding radiotherapy decisions and to establish standardized protocols for their use in clinical settings [84,85].

Hsu et al. further explored the predictive capabilities of CRC organoids by constructing them from normal colon, adenoma, and tumor tissues of patients undergoing neoadjuvant therapy. Their findings indicated that while normal colon tissues retained inherent resistance to radiation, the transition from adenoma to adenocarcinoma was associated with increased radiosensitivity. This highlights a critical phase in tumor progression where treatment responses may vary significantly among patients. Importantly, the study included clinical validation by correlating organoid-based radiosensitivity with actual responses to chemoradiation in rectal cancer patients, strengthening its translational relevance. However, the study was based on a relatively small patient cohort (n = 13 patients), which may limit broader applicability [86].

In another study, Cho et al. investigated the correlation between drug responses of PDOs and clinical responses in matched CRC patients, as well as the progression of their disease. Organoids were established from 54 patients who had not undergone any previous therapy, except for one individual, and were assessed through whole-exome sequencing. To quantify the diverse responses to anticancer treatments observed in these organoids, researchers established an “organoid score”. A higher score was found to be significantly linked to reduced tumor regression rates in patients receiving standard treatment. Furthermore, patients with an organoid score of 2.5 or higher experienced poorer progression-free survival outcomes compared to those with lower scores. However, as the cohort predominantly included treatment-naïve patients, with only one having received prior therapy, the findings may have limited generalizability to pretreated or therapy-resistant CRC populations, introducing a potential selection bias [87]. 

Geevimaan et al. highlights the development of a living biobank of 151 PDOs from 148 CRC patients, including three with matched liver metastases, PDOs from advanced CRC patients to predict responsiveness to oxaliplatin-based chemotherapy. While the incorporation of oxaliplatin with 5-FU has led to improvement in disease-free survival, some patients remain resistant. In this study, 42 PDOs from advanced-stage (III and IV) patients were classified as oxaliplatin-resistant (OR) or oxaliplatin-sensitive (OS) based on their sensitivity assessed in vitro and in PDO-xenografted tumors in mice. Importantly, drug responses from 17 of these PDOs were correlated with the clinical outcomes of patients treated with FOLFOX, providing a level of clinical validation. Molecular profiling of 16 PDOs (8 OR and 8 OS), identified distinct gene signatures for OR and OS PDOs, leading to the discovery of 18 predictive biomarkers for oxaliplatin response. Additionally, candidate drugs that could enhance sensitivity in resistant PDOs were identified, including inhibitors targeting the c-ABL and Notch pathways [88].

Organoids have also been employed to evaluate the response to targeted therapies, like cetuximab, with the findings suggesting that organoid genotypic characteristics can accurately reflect the patient’s clinical response [89]. De Oliveira’s study evaluated the expression of PFKFB3, a glycolysis regulator, and its association with patient outcomes in rectal and colon tumors. Researchers tested the impact of a novel PFKFB3 inhibitor, KAN0438757, on CRC cells and PDOs. Results indicated that KAN0438757 effectively reduced PFKFB3 levels, leading to decreased cancer cell motility, invasion, and survival, while demonstrating significant cytotoxic effects on PDOs without affecting normal colonic organoids. Importantly, KAN0438757 was well tolerated in vivo, showing no systemic toxicity in mice. These findings support the potential of targeting glycolysis as a therapeutic strategy for improving drug sensitivity in CRC treatment. However, the study’s evidence is limited by a small and unspecified number of patient-derived organoids and use of a single in vivo model [90]. Moreover, rectal cancer organoids have been shown to exhibit varying sensitivities to chemotherapy and radiation, mirroring the clinical chemoradiation responses observed in patients [91].

PDOs can also be preserved, establishing a valuable biobank. This allows the categorization of PDOs with the analysis of drug sensitivity profiles across a collection, streamlining comparisons between individual patient responses. The complexity of tumors in treatment lies in their heterogeneity, which can be effectively addressed by leveraging a large organoid biobank and personalized treatment strategies to predict drug responses and patient outcomes [92,93]. The genomic analyses conducted by Van de Wetering et al. revealed that the organoids closely mirror the genomic characteristics of primary colon cancer, demonstrating a high degree of consistency with large-scale mutation profiles observed in CRC. This study involved the successful creation of a biobank comprising 22 PDOs and 19 normal adjacent organoids derived from 20 CRC patients. However, while the biobank represents a significant advance, the relatively small patient cohort may limit generalizability, and the absence of functional validation or clinical outcome correlation highlights the need for further studies to assess predictive value and therapeutic applicability [94]. Similarly, Vlachogiannis et al. developed a comprehensive living CRC organoid biobank using 110 fresh biopsy samples from 71 patients across four prospective phase 1/2 clinical trials, focusing on metastatic, heavily pretreated colorectal and gastroesophageal cancers. Their findings highlighted notable morphological similarities between the organoids and patient tissues, and sequencing data indicated that the phenotypes and genotypes of the organoids were highly representative of the original tumors. In their analysis, they reported 100% sensitivity, 93% specificity, 88% positive predictive value, and 100% negative predictive value in forecasting patient responses to targeted therapies or chemotherapy (Fisher’s exact test, *p* < 0.0001), underscoring the potential of PDOs to simulate cancer behavior ex vivo and support molecular pathology-driven decision-making in early-phase clinical trials [92]. This capability is particularly advantageous for preclinical, high-throughput drug screening, allowing for efficient testing of therapeutic responses without requiring patient participation. Additionally, these organoid biobanks hold promise for applications in gene editing, novel drug development, and regenerative medicine, ultimately facilitating the creation of more individualized treatment strategies for CRC patients (Table 2).

## 4. Microfluidic Platforms for Modeling the Tumor Microenvironment and Drug Responses

Recent advancements in 3D models include the development of microfluidic devices, which enable precise manipulation of small liquid volumes and compartmentalization of various cell types. These devices provide a valuable platform for studying cancer and other diseases, significantly enhancing research on vasculature, cancer progression, metastasis, and applications in drug development and diagnostics [103].

While static organoid cultures have been foundational in advancing organoid research, they present intrinsic limitations due to the absence of dynamic physiological cues. In particular, the lack of fluid flow results in inadequate nutrient and oxygen diffusion, generating concentration gradients that can impair cell viability and differentiation, especially in larger organoids. In contrast, dynamic culture systems, such as microfluidic platforms and perfusion bioreactors, enhance mass transport and introduce shear forces that support improved cellular maturation and tissue organization. These systems enable the sustained growth of organoids at millimeter scales while preserving viability and functional integrity over extended periods, thereby providing a more physiologically relevant in vitro model that better recapitulates in vivo conditions [104,105].

In a 2020 study, researchers demonstrated that the use of microwell array microfluidic devices significantly enhances the investigation of tumor–stroma interactions in 3D cell culture. By enabling precise one-to-one pairing of tumor and fibroblast spheroids, this innovative approach allows for high-content imaging and a detailed analysis of metastatic processes in vitro. The ability to form 240 distinct tumor–stroma pairings within a compact area of 1 cm² addresses the limitations of conventional 3D cultures, particularly regarding heterogeneity and quantitative measurement. The results revealed that tumor spheroids could envelop fibroblast spheroids, illustrating the dynamic interactions that occur during metastasis. Additionally, the researchers found that specific chemicals could effectively modulate these interactions, either promoting or inhibiting the 3D metastatic process. The capacity of this microfluidic system for time-resolved measurements of spheroid merging provides valuable information on the metastatic phenotype between different types of tumors [106].

One of the significant obstacles of traditional 3D models is their inability to incorporate vascular structures, which can compromise their physiological applicability. However, certain microfluidic devices have been developed to address this issue by integrating endothelial cells, effectively mimicking the vasculature in CRC in vivo. This advancement allows researchers to create more accurate representations of the tumor microenvironment, facilitating a better understanding of tumor behavior, angiogenesis, and the interactions between cancer cells and surrounding tissues. By providing a more physiologically relevant model, these microfluidic systems enhance the potential for studying drug responses and disease mechanisms in a controlled environment [107].

In a 2016 study, researchers developed a vascularized microtumor (VMT) platform that represents a significant advancement in microphysiological systems, allowing for the in vitro modeling of human tumors with integrated vascular structures. This “tumor-on-a-chip” approach integrates human tumor and stromal cells within a 3D ECM, relying on perfused microvessels for nutrient delivery. The platform facilitates the progression of colorectal and breast cancer cells, which respond effectively to standard therapies. Notably, it distinguishes between vascular-targeting agents, revealing that while drugs targeting only VEGFRs (e.g., Apatinib and Vandetanib) are ineffective, those targeting VEGFRs, PDGFR, and Tie2 simultaneously (e.g., Linifanib and Cabozantinib) can induce vascular regression. Additionally, imaging techniques have shown metabolic heterogeneity within the VMTs, with many tumors demonstrating a higher free/bound NADH ratio indicative of aerobic glycolysis [108].

The colon-on-a-chip model is a significant advancement in studying CRC and intestinal diseases, offering a microfluidic platform that replicates the physiological and pathological features of the human colon. While these models provide a relevant tool for cancer research and drug screening, several challenges hinder their widespread adoption. Key issues include the need for high-throughput capabilities to facilitate rapid drug discovery, as current low-throughput devices must be adapted to support larger-scale screening. Additionally, the complexity of microfabrication requires specialized skills, limiting consistent use outside specialized research environments. Existing models also struggle to accurately recreate the intricate layers of the colonic wall, which are essential for mimicking true physiological conditions. Recent advancements have shown promise in addressing these challenges, however. For example, colon-on-a-chip platforms can effectively co-culture cancerous colonic epithelial cells with anaerobic bacteria under controlled conditions. Innovations such as 3D printing have led to simpler culture vessels that allow for larger volumes of colorectal cancer microtissues with continuous perfusion, enhancing the relevance of these systems.

## 5. Future Directions and Implications

The increasing incidence of CRC necessitates innovative approaches for early detection and treatment. Current research features the potential of organoids as a transformative tool in CRC management, specifically in understanding tumor biology and personalizing therapy. Even though current organoid-based approaches are producing valuable information relevant to treatment and management of CRC patients, there are several areas in which improvements could be made. For example:

Clinical translation:-Development of specialized CRC-specific organoid models that would include normal colon and polyps from cancer-free individuals and CRC patients. Such organoid models will enable researchers to study the progression from benign lesions to malignant tumors, facilitating the identification of early biomarkers for CRC detection and intervention [25].-Establishment of reproducible long-term cultured CRC organoids. This will shed light on tumor evolution and resistance mechanisms and will also help identify genetic markers associated with therapy response or resistance that facilitate the development of more effective personalized therapies [109]. The utilization of CRC organoids to study drug resistance patterns can reveal critical insights into why certain patients do not respond to standard therapies. This knowledge could guide the development of combination therapies or novel agents targeting resistant phenotypes.-Creation of composite organoids that incorporate various CRC-relevant cell types, including immune cells and stromal components. This will help mimic the TME more accurately, enhancing our understanding of tumor-immune interactions and informing immunotherapy strategies. However, implementing these advanced models presents several challenges. Technically, co-culturing multiple cell types requires careful optimization of media conditions, timing of cell addition, and maintenance protocols to ensure viability and physiological relevance. Moreover, inter-laboratory variability and lack of standardized protocols can limit reproducibility and scalability. To address these barriers, future efforts should focus on the development of standardized co-culture methodologies and shared organoid biobanks. Training programs and collaborative platforms can also help disseminate technical expertise. Integration of automation (e.g., microfluidics, bioprinting) and advanced imaging technologies may reduce manual labor and improve consistency [110,111]. These advancements will be crucial for translating composite organoid models into robust platforms for studying CRC biology and guiding immunotherapy development.

Microbiome modeling:

-Development of organoid-based models to assess the impact of environmental factors, known as exposomes, on CRC, which can provide insights into how diet, microbiota, and other external factors influence cancer development. This approach could lead to preventive strategies tailored to individual risk profiles.

Technological integration:

-Advancements, like automated bioprinting, machine learning integration and rapid data acquisition, in high-throughput screening modalities utilizing organoids will allow for rapid testing of drug efficacy across diverse patient-derived models. This will streamline the process of identifying effective treatments tailored to individual genetic backgrounds.-Technical innovation will need to be made to rapidly integrate multi-omics data—genomics, transcriptomics, proteomics—from organoid cultures. This will yield an inclusive understanding of CRC heterogeneity and treatment responses.

## 6. Potential Limitations

Although organoids have emerged as a powerful tool in biomedical research, they face notable limitations that hinder their full potential. For example:-Present-day CRC organoid systems still cannot accurately replicate the intricate interactions and microenvironments of human organs.-The absence of vascularization within organoids limits their ability to support sustained growth and mimic certain physiological processes. Traditional organoids lack intrinsic vascularization, which limits their ability to support sustained growth, model angiogenesis, or replicate physiological nutrient and oxygen gradients. However, emerging microfluidic technologies and vascularized organoid platforms, such as tumor-on-a-chip models, have begun to address this limitation by incorporating perfused microvessels and endothelial networks. While promising, these systems are still under development and not yet standard in most organoid-based studies.-Issues related to variability and stability during long-term culture pose challenges to reproducibility and reliability in experimental studies.-Difficulty of scaling organoid production, which restricts their application in large-scale research or clinical settings.

Recent advancements have begun to address these limitations. For example, Hachey et al. introduced a vascularized micro-tumor model derived from patient CRC tissues, incorporating endothelial and stromal components to more accurately reflect clinical disease and therapeutic response [112]. Similarly, Lee et al. developed the Vascularized Tissue on Mesh-Assisted Platform (VT-MAP), a novel open microfluidic system that allows for the dynamic co-culture of PDOs with perfusable vasculature. This system not only enhances organoid viability but also enables size-dependent drug testing that closely mirrors in vivo pharmacodynamics [113]. Efforts to address vascularization more broadly include the IFlowPlate system, which enables high-throughput culture of perfusable vascularized colon organoids in a 384-well format [114], and the work by Zhao et al., who discussed the role of ECM components like Matrigel in supporting organoid culture and vascular network formation [115]. Mitrofanova et al. developed bioengineered human colon organoids, termed “mini-colons”, that exhibit in vivo-like cellular complexity and function. These mini-colons incorporate a diverse array of cell types, including mucus-producing goblet cells and mature colonocytes, and maintain long-term tissue integrity without passaging. This advancement enhances the physiological relevance of organoid models, providing a more accurate representation of human colon tissue for research and therapeutic applications [116].

In conclusion, the future of CRC organoid research holds significant promise for enhancing early detection, understanding disease mechanisms, and personalizing treatment strategies. Continued advancements in technology, such as the integration of high-throughput screening techniques and single-cell sequencing technologies, and methodology, like CRISPR, will be essential for translating these findings into clinical practice. This will improve outcomes for patients with colorectal cancer.

## 7. Conclusions

Colorectal cancer remains a major global health challenge, and the integration of organoid technology into CRC research represents a transformative leap in how we model, study, and treat this disease. As 3D organoid systems increasingly reflect the genetic, phenotypic, and functional characteristics of patient tumors, they are becoming essential tools for personalized medicine, drug development, and the study of tumor biology. The development of biobanks, composite organoids, and organoid co-cultures with immune and stromal cells have already enhanced the physiological relevance of these models.

Despite advantages, critical challenges remain, particularly around vascularization, reproducibility, and scalability. Addressing these issues will be key to expanding the application of organoids in large-scale screening and clinical settings. Future innovations such as high-throughput bioprinting, machine learning integration, and multi-omics data fusion will likely accelerate the clinical utility of CRC organoids. With continued interdisciplinary efforts, organoids have the potential to reshape translational cancer research and optimize therapeutic strategies for diverse patient populations.

## Figures and Tables

**Figure 1 cancers-17-02129-f001:**
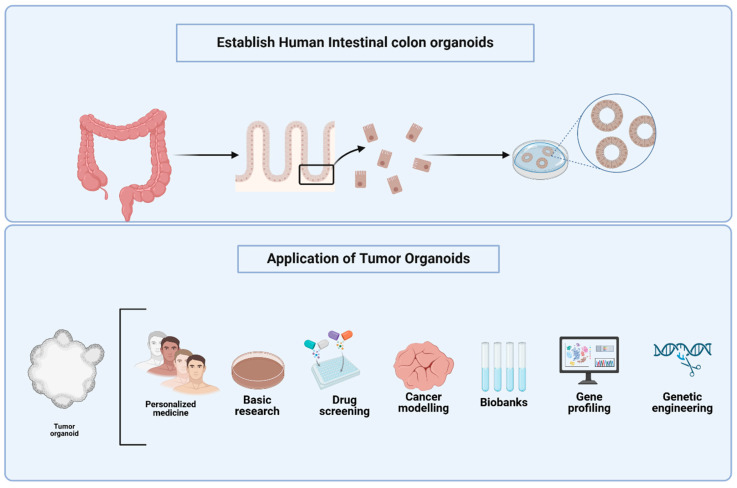
Establishment and Applications of Human Colon Tumor Organoids. The top panel illustrates the sequential process for establishing human intestinal colon organoids, beginning with isolation of crypts from colon tissue and progressing to the generation of three-dimensional colonoids from epithelial fragments. The bottom panel highlights the diverse applications of tumor organoids applications, including their role in personalized medicine, preliminary research studies based on large scale screening, drug screening, cancer modeling, biobanking, gene profiling, and genetic engineering. Tumor organoids are utilized to confirm both histologic and molecular similarity to the original patient tumor, typically assessed using histopathology and high-throughput sequencing approaches such as whole-exome sequencing (WES) and single-cell RNA sequencing (scRNA-seq). For drug screening, organoids are exposed to candidate compounds in standardized protocols that measure cell viability, apoptosis, and transcriptomic response using methods like CellTiter-Glo assays and high-content imaging. These models facilitate the investigation of molecular mechanisms underlying cell fate decisions, cellular interactions, and differentiation, providing a translational bridge between bench research and patient care. Figures were created with BioRender.com.

**Figure 2 cancers-17-02129-f002:**
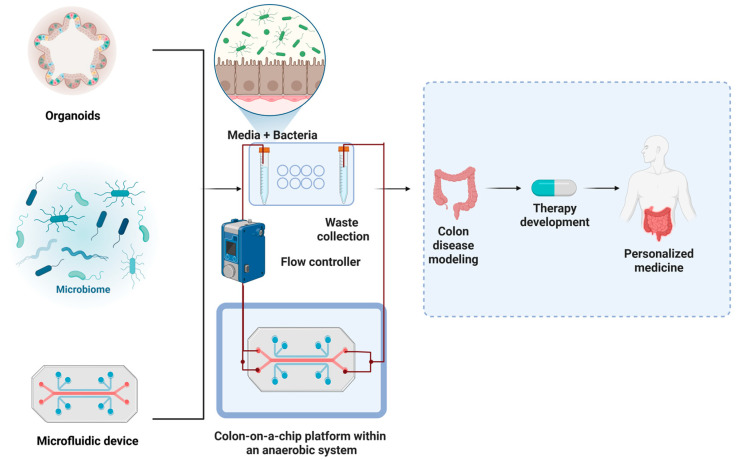
Integrating colon organoids and microorganisms and microfluidic platforms for disease modeling and therapeutic discovery. The colon organoids, when cultivated together with the microbiote and integrated into the microfluid intestine system on a chip, provide a powerful platform for the recapitulation of key aspects of the colon niche. These integrated models enable dynamic analysis of the interaction between host and microorganism, epithelial physiology, and mechanisms specific to disease under almost physiological conditions. Such systems offer unique opportunities to model colon disorders, identify therapeutic targets and advance the precision medicine approach. Figure created with BioRender.com.

**Table 1 cancers-17-02129-t001:** Functions of different bacterial species and their role in the microbiota.

Bacterium	Functions	Microbiota Details	Mechanism	Study Limitations	Ref.
*Clostridioides difficile*	Higher incidence of C. difficile in the CRC patients compared to the healthy individuals	Identified using TaqMan Real-time PCR, significant difference in colonic	Disrupts the gut microbiota and promotes cancer by inducing inflammation	The cross-sectional design restricts the ability to determine causal relationships and does not provide longitudinal data	[47]
*Fusobacterium nucleatum*	Fusobacterium nucleatum promotes inflammatory and anti-apoptotic responses in colorectal cancer cells via ADP-heptose release and ALPK1/TIFA axis activation	Gut bacterial metabolites, dysregulation associated with cancer	Drives tumor progression by triggering inflammation and inhibits cell death	Results from in vitro and animal studies may not accurately reflect the effects observed in humans	[48]
*Streptococcus lutetiensis*	Assisted in diagnosing invasive adenocarcinoma	Bacterial translocation due to disruption of colonic mucus	Promotes cancer by breaching the mucosal barrier	Clinical evidence is limited, and the underlying mechanism has not been fully elucidated in humans	[49]
*Escherichia coli (pks+)*	Production of colibactin, associated with CRC	Prevalence in healthy individuals, tumor development in the colon	Induces DNA damage that leads to cancer	The influence of other genotoxic bacteria cannot be fully ruled out as a confounding factor.	[50]
*Parabacteroides distasonis*	Enhance antitumor immunity by modulating CXCL10 and CD8+ T cells	Found in the gut, modulation of immune response to colon tumors	Stimulates immune cells to act against tumor growth	Results from animal studies may not directly translate to human colorectal cancer	[51]
*Various gut Firmicutes*	Effects of hypsunation in myeloid cells on colitis and cancer	Increased levels of DHPS and EIF5A Hyp in cells infiltrating the colon in Crohn’s disease patients	Alters the gut microbial ecosystem, disrupting balance and facilitating cancer development	Observational data cannot establish cause-and-effect relationships	[52]
*Various gut microbiota*	Effects of hypsunation in myeloid cells on colitis and cancer	Increased levels of DHPS and EIF5A Hyp in cells infiltrating the colon in Crohn’s disease patients	Alters immune cells activity to regulate inflammation and cancer	Functional significance in human patients not fully validated	[53]
*Fusobacterium nucleatum*	Promotes CRC progression via quorum sensing signaling	Interacts with host hormones, novel strategy for managing pathogenic	Enhances tumor growth by cell-to-cell signaling and hormone interaction	Mechanistic details of host-hormone interactions require further study	[54]
*Streptococcus gallolyticus*	Induces inflammatory responses, enhances carcinogenesis	Commonly found in CRC patients, associated with advanced disease	Activates inflammation that promotes tumor initiation and progression	The role of infection as a direct etiological factor in cancer is still undetermined	[55]

**Table 2 cancers-17-02129-t002:** Metabolic Effects of Drugs in Colorectal Cancer (CRC) and Their Therapeutic effect.

Drug	Target	Metabolic Effect in CRC	Therapeutic Outcome	Ref.
RO5126766	RAF-MEK-ERK signaling pathway	Decreased GLUT1 expression, which results in lower glucose uptake	Reduces CRC xenograft growth	[95]
IDF-11774 (LW6)	hypoxia-inducible factor-1 (HIF-1) inhibitor	Upregulation of HIF-1α correlates with diminished glucose uptake, suppressed glycolysis, and ATP depletion in HCT116 cells	Suppression of HCT116 xenograft growth	[96]
Wogonin	Inhibiting PI3k/Akt signaling pathway	Hypoxia-induced inhibition of HIF-1α, glucose consumption, and lactate synthesis in HCT116 cells	Inhibition of HCT116 xenograft growth	[97]
tephrosin with 2-deoxy-D-glucose drug combination	HT-29 and SW-620	TSN and 2-DG synergistically promoted intracellular ATP depletion and robust AMPK activation, ultimately suppressing the mTOR pathway	The addition of TSN to 2-DG exacerbated intracellular ATP depletion and prevented 2-DG-induced autophagy by inhibiting the activation of eEF-2K, leading to an increase in apoptosis	[98]
3-bromopyruvate (3-BP)	Hexokinase II (HK-II)	Decreased ATP levels in SW480 and HT29 cell lines	3BP promotes various forms of cell death through energy depletion in vitro, reducing resistance to drug-induced cell death. Its anti-tumor activity in vivo shows its potential as a therapeutic option for CRC	[99]
PFK-15 (1-(4-pyridinyl)-3-(2-quinolinyl)-2-propen-1-one)	PFKFB3 (6-phosphofructo-2-kinase)	Inhibition of oxaliplatin-induced autophagy	Increased sensitivity of SW480 cells to oxaliplatin	[100]
WZB117	Glucose Transporter 1 (GLUT1)	Induction of platelet-derived growth factor	Increased levels of glycolysis, resulting in higher intracellular lactate and acidic byproduct accumulation	[101]
Oridonin	Active diterpenoid	It deactivates phosphorylated AMPK and downregulates the GLUT1 and MCT-1 expression.	Suppresses glucose uptake, decreases lactate production, and triggers autophagy and cell mortality in CRC cells.	[102]

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
