# Peer review of "The Role of Organoids in Advancing Colorectal Cancer Research: Insights and Future Directions"

_cancers, 2025, doi:10.3390/cancers17132129_

Round 1

Reviewer 1 Report (Previous Reviewer 3)

Comments and Suggestions for Authors

The revised manuscript shows significant improvement. I have no further comments or concerns and recommend accepting the paper in its current form.

Reviewer 2 Report (Previous Reviewer 1)

Comments and Suggestions for Authors

Thank you for the opportunity to review the revised manuscript. 

The authors have addressed almost all concerns, and the manuscript is now significantly improved in clarity, structure, and relevance so I believe it is suitable for publication in Cancers without further revisions.

This manuscript is a resubmission of an earlier submission. The following is a list of the peer review reports and author responses from that submission.

Round 1

Reviewer 1 Report

Comments and Suggestions for Authors

Peer Review – Cancers (MDPI)

Manuscript Title: The Role of Organoids in Advancing Colorectal Cancer Research: Insights and Future Directions

Recommendation: Major Revision

Overall Comment

This manuscript presents a thorough and timely review of the utility of patient-derived organoids (PDOs) in colorectal cancer (CRC) research. The authors comprehensively cover the current applications of organoid models—from tumor biology and drug screening to immune co-culture systems and microbiome integration. The breadth of content is commendable.

However, the manuscript is currently too descriptive in nature. It often summarizes studies without critical evaluation or synthesis. Some parts are redundant, others contain conceptual contradictions, and several scientific claims need clarification. With substantial improvements in organization, clarity, and analytical depth, this paper has the potential to become a valuable resource for CRC researchers.

Major Comments

  • Redundancy and Overlength: The sections on co-culture models and microbiota (pages 5–8) are repetitive and overly detailed. Streamlining with thematic grouping would improve readability.
  • Lack of Critical Synthesis: Rather than summarizing study outcomes sequentially, the review should synthesize key findings, highlight limitations, and draw comparisons across methods (e.g., direct vs. indirect co-cultures).
  • Microbiota Section is Disconnected: Although microbiome-organoid interactions are important, this section feels disconnected from the rest of the manuscript. Reframe or reposition it earlier, with clear relevance to CRC progression or treatment.
  • Figure Clarification Needed: Figure 1 and 2 need clearer legends. All abbreviations (e.g., CAFs, ECM) should be defined, and figure claims aligned with the text.
  • “Future Directions” Section Needs Structure: Good points are presented, but they are scattered. Grouping into subcategories like “Technological Integration,” “Clinical Translation,” and “Microbiome Modeling” will aid clarity.
  • Language and Style: Revise verbose or awkward sentences (e.g., line 69). Avoid generic terms like “novel insights” and specify what exactly is novel.

Minor Comments (Professional & Insightful)

  • Line 87: Add citation for the foundational Sato et al., Nature 2009 paper on organoid development.
  • Lines 197–198: Explain how ZEB1/Vimentin upregulation relates to CRC invasion or chemoresistance. Add citation if possible.
  • Line 245: The SIRT1–CXCR4/CXCL12 axis is interesting; briefly explain its immunological relevance.
  • Line 248: Discuss how MSI organoid-T cell co-cultures model checkpoint blockade therapy outcomes.
  • Lines 321–322: Use of antibiotics in microinjection systems may affect epithelial viability—please clarify or cite.
  • Line 433: Add citation for the metabolic role of lactate in CSC maintenance.
  • Table 1: Ensure consistency in describing microbial functions and outcomes. Add a column for CRC mechanisms (e.g., inflammation, DNA damage).
  • Line 314 vs. Line 294: Contradiction between Figure 2 claims and earlier discussion of oxygen incompatibility with microbiota co-culture.

Scientific Inconsistencies or Conflicting Claims

  • NK Cell Resistance (Lines 217–227): The CRC organoids resist NK cell-induced apoptosis, yet no mechanism is proposed. This contradicts known sensitivity of MSI-high CRCs to IFN-γ.
  • Genetic Fidelity of PDOs (Lines 159–162): The claim of high mutation fidelity conflicts with the note that none of the preserved mutations are actionable. Clarify how useful PDOs are in precision oncology.
  • CAF Function Contradictions (Lines 118 vs. 198): Early CAFs are portrayed as pro-tumorigenic, while later sections show them inducing EMT (potentially tumor-suppressive in some contexts). Clarify fibroblast subtypes.
  • PDOs Predicting Both Radio- and Chemoresistance: It’s unclear how reliably PDOs predict multiple treatment outcomes (lines 444 vs. 438). Discuss limitations in predictive accuracy across modalities.
  • Vascularization Conflict: Vascularization is claimed to be missing from organoids (line 541), but earlier (lines 374–384) microfluidic vascular models were described. Clarify current capabilities.

With revision and sharper focus, this manuscript can become a foundational reference in the growing field of CRC organoid research.

Reviewer 2 Report

Comments and Suggestions for Authors

Overview:
This manuscript provides a thorough review of colorectal cancer (CRC) organoids. "Organoids" refer to 3D, organ-like cellular structures that closely resemble their native organs. Organoids can be derived from normal adult tissue, cancer tissue, or differentiated pluripotent stem cells. As highlighted in the manuscript, organoids offer advantages over traditional cell culture methods by providing a 3D structure, supporting multiple cell types, allowing easy observation, enabling the addition of other cell types, and integrating with microfluidic technologies.

Comments:
Addressing the comments would significantly improve the manuscript's readability.

  • The headings for Sections 1 and 2 should be more meaningful, with content that clearly reflects the headings.
  • The first three headings are redundant. The title already conveys that the manuscript focuses on using organoids to study CRC, so repeating this in the section headings is unnecessary.
  • The Introduction provides a suitable background on CRC and highlights the limitations of current research techniques. Section 2, which elaborates on the introduction of organoids, could be merged into the Introduction. Currently, this section feels unfocused. Combining and refining these sections would result in a more concise and well-organized introduction to organoids in cancer research.
  • Section 3 introduces patient-specific (personalized) organoid studies. However, since many sections describe patient-derived organoid studies, the focus of Section 3 is unclear. Clarifying whether this section is intended to contrast patient-derived organoids with (1) animal cells, (2) pluripotent cell-derived organoids, and (3) healthy adult tissue would strengthen it. Additionally, while multiple patient-derived organoids enable population-level studies and exploration of heterogeneity, the summaries of the studies presented do not consistently emphasize the unique advantages of using patient-derived organoids (PDOs).
  • Sections 4 and 5 introduce organoid variations used for advanced studies and are clearly written. Highlighting the disadvantages of static organoid culture and the improvements brought by media flow systems is a strong addition.
  • Section 6 again covers personalized medicine, overlapping with Section 3. Creating a new heading for Section 3 or making Section 6 a subsection of Section 3 would improve focus and organization.

Reviewer 3 Report

Comments and Suggestions for Authors

This review article examines the uses and obstacles of colorectal cancer (CRC) organoids, focusing on their importance in modeling tumor biology, drug screening, and personalized medicine. Derived from patient tissues, CRC organoids capture tumor heterogeneity and interactions within the tumor microenvironment (TME), presenting benefits over 2D cultures and patient-derived xenografts. Cited studies indicate that organoids are effective in high-throughput drug screening, achieving 71-90% success rates in preserving genetic integrity to the original tumors. Co-culture systems incorporating cancer-associated fibroblasts and immune cells unveil mechanisms of chemoresistance and immune evasion. Advanced technologies such as microfluidics and bioprinting improve organoid capabilities by simulating vascularization and host-microbe interactions. Despite their potential, challenges persist, including issues related to scalability, absence of vascularization, and variability in long-term culture. These insights underscore the transformative impact of organoids on CRC research.

The review delivers a thorough overview of CRC organoids, effectively emphasizing their benefits compared to traditional models and their roles in drug screening, TME modeling, and personalized medicine. The incorporation of recent studies (up to 2024) and innovative techniques like microfluidics enhances its relevance. Nevertheless, the manuscript necessitates significant revisions to improve scientific rigor, clarity, and comprehensiveness. Major concerns involve an inadequate literature review, lack of critical evaluation of referenced studies, inconsistent terminology, and a brief discussion of methodological challenges.

Comments for authors

Comment 1: The introduction outlines 2D culture and PDX limitations but lacks a review of recent CRC organoid advancements since 2022. It should include systematic reviews from 2023-2024 on organoid applications in CRC to contextualize the review's scope.

Comment 2. The introduction briefly mentions the potential of organoids but does not quantify their impact (e.g., success rates or clinical translation rates) compared to other models.

Comment 3. Line 156-165: The review cites a 71% success rate for PDOs but lacks discussion on variability and influencing factors. Include a comparative analysis of success rates and discuss technical challenges in organoid establishment in this section.

Comment 5: The conversation on co-culture systems emphasizes CAFs and immune cells, yet it overlooks the variability in co-culture protocols, such as direct versus indirect approaches.

Comment 6: Revise Figure 1 to include detailed labels or a legend clarifying key steps like sequencing methods and drug screening protocols

Comment 7. Revise Figure 2 with annotations detailing microfluidic platform specifications or a caption explaining their relevance.

Comment 8. Revise Table 1 to include a column for study limitations.

Comment 9. The “Future Directions” section proposes composite organoids without addressing feasibility or barriers (e.g., cost, technical expertise). Discuss practical challenges and potential solutions for implementing these models.

Comment 10. The review mentions limitations like lack of vascularization but does not cite studies addressing these issues (e.g., vascularized organoid models).

Comment 11. The discussion of personalized medicine cites multiple studies but fails to critically evaluate their methodological rigor (e.g., sample sizes, validation). I recommend assessing the evidence quality and discussing biases in these studies.

Comment 12. The manuscript contains several grammatical errors, awkward phrasing, and inconsistent terminology that reduce readability. For example, “facilitate with developing” (Line 140-141) are grammatically incorrect (should be “facilitate the development of”). Additionally, inconsistent use of terms like “PDOs” and “tumor organoids” creates confusion. I recommend a thorough English language review to correct typos, standardize terminology (e.g., consistently use “patient-derived organoids” or “PDOs”), and simplify complex sentences. Ensuring proper grammar and clear phrasing will enhance accessibility for a broad readership.

Comments on the Quality of English Language

The manuscript contains several grammatical errors, awkward phrasing, and inconsistent terminology that reduce readability. For example, “facilitate with developing” (Line 140-141) are grammatically incorrect (should be “facilitate the development of”). Additionally, inconsistent use of terms like “PDOs” and “tumor organoids” creates confusion. I recommend a thorough English language review to correct typos, standardize terminology (e.g., consistently use “patient-derived organoids” or “PDOs”), and simplify complex sentences. Ensuring proper grammar and clear phrasing will enhance accessibility for a broad readership.